# Transcriptomic-Proteomic Correlation in the Predation-Evoked Venom of the Cone Snail, *Conus imperialis*

**DOI:** 10.3390/md17030177

**Published:** 2019-03-19

**Authors:** Ai-Hua Jin, Sébastien Dutertre, Mriga Dutt, Vincent Lavergne, Alun Jones, Richard J. Lewis, Paul F. Alewood

**Affiliations:** 1Institute for Molecular Bioscience, The University of Queensland, St Lucia, QLD 4072, Australia; a.jin@imb.uq.edu.au (A.-H.J.); m.dutt@uq.edu.au (M.D.); a.jones@imb.uq.edu.au (A.J.); r.lewis@imb.uq.edu.au (R.J.L.); 2Institut des Biomolécules Max Mousseron, Département des acides amines, Peptides et Protéines, Unité Mixte de Recherche 5247, Université Montpellier 2—Centre Nationale de la Recherche Scientifique, Place Eugène Bataillon, 34095 Montpellier CEDEX 5, France; sebastien.dutertre@umontpellier.fr; 3Léon Bérard Cancer Center, 28 rue Laennec, 69008 Lyon, France; vincent.lavergne@protonmail.com

**Keywords:** conopeptide, conotoxin, mass spectrometry, venom transcriptome, 454 sequencing, iTRAQ, proteomics, transcriptomics, cone snail venom, *Conus imperialis*

## Abstract

Individual variation in animal venom has been linked to geographical location, feeding habit, season, size, and gender. Uniquely, cone snails possess the remarkable ability to change venom composition in response to predatory or defensive stimuli. To date, correlations between the venom gland transcriptome and proteome within and between individual cone snails have not been reported. In this study, we use 454 pyrosequencing and mass spectrometry to decipher the transcriptomes and proteomes of the venom gland and corresponding predation-evoked venom of two specimens of *Conus imperialis*. Transcriptomic analyses revealed 17 conotoxin gene superfamilies common to both animals, including 5 novel superfamilies and two novel cysteine frameworks. While highly expressed transcripts were common to both specimens, variation of moderately and weakly expressed precursor sequences was surprisingly diverse, with one specimen expressing two unique gene superfamilies and consistently producing more paralogs within each conotoxin gene superfamily. Using a quantitative labelling method, conotoxin variability was compared quantitatively, with highly expressed peptides showing a strong correlation between transcription and translation, whereas peptides expressed at lower levels showed a poor correlation. These results suggest that major transcripts are subject to stabilizing selection, while minor transcripts are subject to diversifying selection.

## 1. Introduction

Venom variation has been described for secretions produced by various venomous animals, such as cone snails [1], scorpions [2,3], snakes [4,5], spiders [6,7], fire ants [8], and parasitic wasps [9]. The contributing factors leading to such variation comprise geographical location, feeding habits, season, size, and gender [7,10,11,12,13]. Cone snails are marine molluscs that produce complex venom to defend against predators and to capture prey [14]. Over the last decades, the occurrence of individual variation of the dissected and injected venoms of cone snails has been well documented using proteomic approaches [15,16]. Further complexity was added when we discovered that cone snails possess a remarkable ability to inject venoms with differing components in response to predatory or defensive stimuli [17]. To date, such variation has only been examined at the whole venom gland by proteomics approaches [15,16], whereas modulation of mRNA transcripts in the venom gland has not been investigated as a contributor to toxin variability. While evidence for a direct link between mRNA levels in the venom gland and peptide levels in the venom has not been demonstrated, it is generally accepted that mRNA abundance grossly correlates with protein concentrations (correlation coefficient ranging between 0.4 to 0.6) and combinations of post-transcriptional and post-translational regulations have been invoked to account for the remaining ~50% observed variation [18]. With the advent of high throughput sequencing technology and improvement in the sensitivity and throughput of mass spectrometry, we can now directly address these issues. Recently, using an integrated venomics approach, we revealed the occurrence of transcriptomic messiness and variable peptide processing in various cone snail species that may contribute to the venom variation [19].

In this study, we compare, for the first time, transcriptomic and proteomic venom expression profiles of two adult *Conus imperialis* specimens, collected from the same location. *C. imperialis* is a worm-hunting species that feeds exclusively on a group of polychaete marine worms known as “fireworms” (Amphinomidae). Significant toxin variation was revealed at the transcriptomic sequence level with broad overlap between superfamilies and frameworks. Although variation was observed at both mRNA and peptide levels, a core of highly transcribed toxins was consistently detected in the injected venoms of both specimens. Interestingly, sequences with high cDNA read levels correlated well with the levels of peptides translated, whereas sequences with low level reads did not.

## 2. Results

### 2.1. Transcriptomic and Bioinformatic Data Analysis

The 454 pyrosequencing run on the Roche GS FLX Titanium sequencer (1/4 of a plate equivalent for each of the two specimens of *C. imperialis*) generated 220,516 (Specimen 1, S1) and 332,390 (Specimen 2, S2) cDNA reads after trimming and removal of low-quality sequences. Newbler and Trinity assembled de-multiplexed read sequences that produced contigs with consistent length ranges, N50, N75, and N90 values (Appendix A). The average length after assembly was not much different compared to the raw reads, e.g., the trimmed raw reads average length ~390 bp, after the Trinity assembly ~480 bp, and after Newbler assembly 430–460 bp.

Sorting and classification into gene superfamilies of the raw cDNA reads was performed using ConoSorter [20]. After motif searching using parameters generated from the ConoServer database [21], a total of 267 unique conopeptide precursors were retrieved, including 96 from the specimen S1 and 233 from specimen S2, with 62 overlapping (Figure 1A and Appendix A). Assembled data sets were also subjected to the same conotoxin sequence analysis, however, the number of conopeptide sequences identified from the assembled contigs with ConoSorter remained low compared to the direct analysis of the cDNA reads. From the 62 common sequences initially retrieved from the cDNA read datasets of both specimens, only 15 (Newbler) and 18 (Trinity) presented after the assembly. Whereas the low expressed sequences have been eliminated during the assembly as expected, it was quite surprising that some highly expressed sequences were absent after the assembly (Appendix A). Indeed, most algorithms are designed to reduce substitutions, deletions, and insertions events, which, in the case of hypervariable genes such as conotoxins, will likely eliminate some minor yet important true biological variations [22]. Comparison of conopeptide sequences measured at the cDNA level allows for a clear distinction between high and low levels of expressed transcripts. Therefore, this work from now on was carried out using the reads, not contigs, as the short sequences of conotoxin precursors (70–100 a.a. on average) have been proven fully covered by most 454 reads (average length 390 bp).

Using a 75% signal peptide sequence identity cut-off [23,24], 17 superfamilies were found in both specimens and 2 superfamilies (S and putative SF-im6) were identified only in S2 (Table 1 and Figure 1B).

Among the 17 common gene superfamilies, 12 were already described (i.e., A, D, E, I1, I2, K, M, N, O1, O2, P and T) and five were novel (coded SF-im1-5). The signal peptides and cysteine frameworks of the superfamilies are listed in Table 1. SF-im1 is not closely related to any known superfamily (45% M sequence identity) and contains only two cysteine residues in the predicted mature region. Both SF-im2 and SF-im5 are related to superfamily O3 (56% and 54% sequence identity, respectively), though they appear clearly different from each other. Superfamily SF-im3 is related to superfamily D (56%) and SF-im4 is related to superfamily I2 (58%). In addition, superfamily S was uniquely found in specimen 2, along with another new superfamily (SF-im6, 40% I3). Overall, the six most abundant superfamilies in the venom of *C. imperialis* are A, D, T, I2, O2, and K. The total number of paralogs and cDNA reads for each superfamily are listed and compared in Table 2. 

### 2.2. Transcriptomic Intraspecific Variation

Striking variation was observed at the transcriptomic level. Indeed, many more paralogs were found for almost every superfamily in S2 than in S1 (Figure 1B). Further, specimen S2 produced more than 1.5 times as many cDNA reads as that of S1. Whereas the top 2–3 ranking paralogs within each superfamily (based on read frequency) appear remarkably conserved between the two individuals, there are some notable exceptions. First, there are cases where the ranking order for the lowest to the most highly expressed sequence is changed, as follows: Within the superfamily M, the 2nd (41 reads), 3rd (19), and 4th (5) most highly expressed paralogs in S1, become the 3rd (25), 2nd (43), and 5th (5) in S2, respectively (Appendix A). More striking, some well-known sequences (e.g., α-conotoxin ImI [25] and Lys conopressin-G [26]), though present within S2 at low abundance (1 read), are undetectable in S1. In fact, from the well-characterized conotoxins in *C. imperialis* venom, only ImIIA is found to be abundant in these two specimens (>200 reads). The PCR amplified product α-conotoxin ImII [27], on the other hand, is missing in both individuals.

### 2.3. Proteomic Intraspecific Variation

The predation-evoked venoms of the two same specimens of *C. imperialis* were collected 3 days prior to mRNA extraction and analyzed using standard LC-ESI-MS and the mass lists were extracted and compared using dedicated bioinformatic tools. When overlaid, the TIC profiles appear grossly similar between individuals (Figure 2A) due to the presence of ~10–15 abundant peptides (Figure 2B). However, when all extracted masses are taken into consideration, there is only 30%–50% overlap between these two animals (232 masses) (Figure 2A). Interestingly, although the venom gland transcriptome of S2 generated more sequences, it produced less complex predation-evoked venom than S1 (476 masses versus 645 masses). 

### 2.4. Transcription and Translation

To determine if the relationship between transcription and translation contributed to venom variation, the T superfamily was analyzed in detail, as it provided a convenient case study. Indeed, the precursor of ImVC (T-*S1*), the second most highly expressed superfamily T peptide (1570.66 Da; FLNTICCWSRACCG-NH_2_), was found in the venom gland transcriptome of S1, but not in S2, and this peptide was relatively abundant in the predation-evoked venom of S1 but remarkably absent from the corresponding venom of S2. Quantitatively, the transcriptomic ratio between ImVC and the most highly expressed superfamily T peptide (Im5.4) is 1:12 and the peptide precursor intensity ratio is 1:4, with the difference of the ratio for tandem mass spectrometry (MSMS) identification likely due to the stronger ionization of an arginine residue instead of a glycine at position 10 (see Table 3).

### 2.5. Correlation between mRNA and Peptide Levels

To quantitatively determine the relationship between mRNA and peptide levels, we employed an iTRAQ labelling proteomic strategy involving tandem LC-ESI-MSMS to compare the relative abundance of conotoxins versus their cDNA read number. The injected venom was reduced and alkylated prior to tryptic digestion [28] and 8-plex labelled samples were pooled and analyzed. A control run without iTRAQ labelling was analyzed for each of the milked samples with or without alkylation and tryptic digestion. Figure 3 shows the comparison of the ratio of iTRAQ-detected conopeptides (peptide level) and the number of cDNA reads (mRNA level) between the two animals. In general, highly expressed peptides show a good correlation between mRNA and peptide level abundance, whereas peptides expressed at low levels do not. Further details relating to the conotoxin precursors, mature peptides, and cysteine frameworks are given below.

### 2.6. Representative Conotoxin Precursors, Mature Peptides, and Cysteine Frameworks by Gene Superfamily

#### 2.6.1. Superfamily A

The number of cDNA reads and conotoxin variants were most abundant in this superfamily, with a total of 5826 cDNA reads and 52 paralogs identified. A_1 and A_2 (Table 4) were the two most abundant sequences, which both contained only a single cysteine residue in their sequences and likely represent a new pharmacological class. Surprisingly, despite their high read number neither peptide was found in the proteome. In contrast, one typical α-conotoxin (A_3, known as ImIIA/Bn1.3) was identified with the classic framework I (CC-C-C). MSMS identified A_3 (ImIIA) in the venom with a common amidated C-termini and one hydroxyproline (Appendix A). 

#### 2.6.2. Superfamily D

Superfamily D had a total 2421 cDNA reads and 29 paralogs were identified. The highly abundant D_1 (Table 4) had previously been observed in the venom of *C. litteratus* (cDNA work, Lt15.5) [21] and was highly expressed in both specimens (438 reads, S1 and 1856 reads, S2). The predicted mature peptide had a unique framework (C-C-C-CC-C-C-C-C-C) that differed from other D-superfamily members with framework XX (C-CC-C-CC-C-C-C-C) [29]. MSMS analysis confidently identified D_1 after tryptic digestion. The ratio of peptide intensity (S1:S2 = 1:4.8) correlated well with the ratio of mRNA (S1:S2 = 1:4.2) between the two specimens (Figure 3A).

#### 2.6.3. Superfamily E

Superfamily E, recently identified in *C. marmoreus* at very low abundance [23], was found to be relatively abundant in the venom of *C. imperialis*, with a total of 204 reads and 3 paralogs identified. Interestingly, E_1 (Table 4) contained no propeptide region and the mature peptide with framework XXII (C-C-C-C-C-C-C-C) was directly cleaved off the signal sequence after a SignalP predicted L--K cleavage site. MSMS analysis confirmed this sequence at the peptide level. The ratio of peptide intensity (S1:S2 = 1:2.0) correlated well with the ratio of mRNA (S1:S2 = 1:1.6) between the two specimens (Figure 3A).

#### 2.6.4. Superfamily I1

The I1 superfamily had only 55 cDNA reads and 3 paralogs identified. Conopeptides from this superfamily have been found to induce an excitatory effect when injected into mice [30]. I1_1 (Table 4) was transcribed in both specimens at relatively low levels (cDNA reads 6 and 28). The mature peptide had 34 amino acids and 4 disulfide bonds (framework XI, C-C-CC-CC-C-C). MSMS showed that this sequence was translated in both animals, although the ratio at peptide level (S1:S2 = 1:1.0) did not correlate well with the cDNA level (S1:S2 = 1:4.7) (Figure 3B).

#### 2.6.5. Superfamily I2

Compared to the I1 superfamily, I2 was highly abundant with a total of 1426 cDNA reads and 10 paralogs identified. As expected for I2 members, the predicted mature peptides were placed between the signal peptide and the propeptide regions. The cysteine framework was the same as in the I1 superfamily (XI, C-C-CC-CC-C-C). I2_1 (Table 4) was identified with 141 S1 and 740 S2 cDNA reads. The ratio of peptide intensity (S1:S2 = 1:5.4) correlated well with the ratio of mRNA (S1:S2 = 1:5.2) between the two specimens (Figure 3A). I2_2 was also identified at the peptide level (Appendix A). 

#### 2.6.6. Superfamily K

The K superfamily has recently been characterized at the peptide level and the mature conotoxin shows a double helical structure within framework XXIII (C-C-C-CC-C) [31]. The K superfamily is abundant in the venom of *C. imperialis*, with a total of 1032 reads and 22 paralogs identified. K_1 (known as Im23b) was identified with 155 S1 and 348 S2 cDNA reads. The ratio of peptide intensity (S1:S2 = 1:2.0) correlated well with the ratio of mRNA (S1:S2 = 1:2.2) between the two specimens (Figure 3A). K_2 (known as Im23a) was also identified at the proteomic level though no correlation was determined as the sequence identified for quantitation overlaid with K_1. K_3 is another relatively abundant paralog with 60 S1 and 109 S2 cDNA reads. The ratio of peptide intensity (S1:S2 = 1:1.4) correlated well with the ratio of mRNA (S1:S2 = 1:1.8) between the two specimens (Figure 3A).

#### 2.6.7. Superfamily M

Superfamily M comprised a moderate proportion of total conotoxin transcripts, with a total of 498 cDNA reads and 16 paralogs identified. The most abundant, M_1(Table 4), was identified with 238 S1 and 343 S2 cDNA reads and it contained an obvious pre-sequence cleavage site (KR), resulting in a short mature peptide of 14 amino acids and a cysteine framework III (CC-C-C-CC). The ratio of peptide intensity (S1:S2 = 1:1.4) correlated well with the ratio of mRNA (S1:S2 = 1:1.4) between the two specimens (Figure 3A). Two conopeptides M_2 (known as Eu3.3) and M_3 (known as Bt3.1) were also identified in both transcriptomes. Although they had 20–40 cDNA reads, their mature sequences were not detected in the milked venoms. M_4 contained an RR cleavage site and cysteine framework IX (C-C-C-C-C-C) and was identified by only 2 S1 and 9 S2 cDNA reads. Although the mature sequence for M_4 was confirmed by MSMS analysis of both specimens, the ratio at peptide level (S1:S2 = 1:1.9) did not correlate well with the cDNA level (S1:S2 = 1:4.5) (Figure 3B).

#### 2.6.8. Superfamily N

Superfamily N, first been discovered in *C. marmoreus* venom, was found in the venom of *C. imperialis* in low abundance, with a total of 34 reads and 2 paralogs identified. N_1 was identified with 16 S1 and 6 S2 cDNA reads and contained an L–R cleavage site. The predicted mature peptide contained a novel framework (C-C-CC-CC-C-C-C-C). MSMS analysis indicated that this sequence was translated in both specimens, although the ratio at peptide level (S1:S2 = 1:3.6) did not correlate with the cDNA level (S1:S2 = 1:0.4) (Figure 3B).

#### 2.6.9. Superfamily O1

Surprisingly, though O1 is one of the most widespread superfamily in cones and found in early diverging taxa such as *C. californicus* [32], the number of cDNA reads and conotoxin variants were low in both transcriptomes, with a total of only 76 cDNA reads and 6 paralogs found. O1_1 (known as conotoxin-3) contained a pre-sequence cleavage site (KR), resulting in the predicted mature peptides containing 30 amino acids and a Type VI/VII (C-C-CC-C-C) framework (Table 4). No MSMS evidence was found for this sequence in the milked venom.

#### 2.6.10. Superfamily O2

Compared to O1, superfamily O2 was much more abundant with a total of 1038 reads and 25 paralogs found. No quantitation of peptide segments could be retrieved by iTRAQ from this superfamily, although two paralogs O2_1 (known as Im6.2) and O2_3 with a VI/VII (C-C-CC-C-C) framework were identified at the peptide level (Table 4, Appendix A). 

#### 2.6.11. Superfamily P

A total of 400 cDNA reads and 18 paralogs were identified for this superfamily. No quantitation of peptide segments could be identified by iTRAQ from this superfamily. Although three paralogs (P_1-P_3) (Table 4) were identified with a RK cleavage site and cysteine framework IX (C-C-C-C-C-C), only P_3 was confidently identified at the peptide level by MSMS (Appendix A). 

#### 2.6.12. Superfamily S

Superfamily S was only found in S2 with total reads of 19 and 2 paralogs identified. Paralog S_1 (Table 4) contained a signal peptide with 80% sequence identity to the classic S superfamily. S_1 was identified in S2 with moderate cDNA reads (S1, 0 and S2, 10) and contained a pre-sequence cleavage site (R-R) that resulted in a predicted mature peptide of 37 amino acids and 4 disulfide bonds with a Type VIII (C-C-C-C-C-C-C-C) framework. No MSMS evidence was found for this sequence in the milked venom.

#### 2.6.13. Superfamily T

Superfamily T was the third most abundant superfamily behind A and O2, with total reads of 2166 and 46 paralogs identified. Surprisingly given the high transcriptomic expression levels, iTRAQ was unable to quantify this peptide. Paralogs T_1 (known as Im5.4) and T_2 both have an elongated N-terminal tail and a classic T superfamily cysteine framework V (CC-CC). The sequence of T_1 was identified confidently at the peptide level by MSMS (Appendix A).

#### 2.6.14. Superfamily SF-im1

Whereas this novel signal peptide has some similarity to that of the superfamily M (45% homologous), the mature sequences differ greatly with a different cysteine framework. The total number of reads for this superfamily was 85 for the 10 paralogs identified. The full precursor of SF-im1_1 was identified with 7 S1 and 25 S2 cDNA reads. The precursor was long and contained a pre-sequence cleavage site (RR) and a post-cleavage site (KK), resulting in predicted mature peptides of 27 amino acids containing only a single disulfide bond. The mature peptide was identified by MSMS after tryptic digestion (Appendix A).

#### 2.6.15. Superfamily SF-im2

This novel superfamily contained a signal peptide distantly related to the superfamily O3 (56% sequence identity). The total reads for this superfamily were 81 with 10 paralogs identified. SF-im2_1 was identified with 494 S1 and 1027 S2 cDNA reads but could not be quantified by iTRAQ. It contained a novel cysteine framework with 8 cysteine residues (C-C-C-C-C-C-CC) and predicted pre-cleavage sites after KR that would generate mature peptides of 35 amino acids. The mature peptide was validated by MSMS after tryptic digestion (Appendix A).

#### 2.6.16. Superfamily SF-im3

This superfamily contained a signal peptide, with 56% sequence identity to the D superfamily, that was distinct from SF-im1 and SF-im2 signal peptides and their corresponding cysteine frameworks. There were 73 cDNA reads and 6 paralogs identified. SF-im3_1 was identified with moderate cDNA reads (S1, 4 and S2, 55). It contained a pre-sequence cleavage site (KR) that resulted in a predicted mature peptide of 50 amino acids and 5 disulfide bonds with a classic XX (C-CC-C-CC-C-C-C-C) framework. No MSMS evidence was found for this sequence.

#### 2.6.17. Superfamily SF-im4

The total read numbers for this superfamily were 37 for the 2 paralogs identified. Similar to superfamily I2 (58% sequence identity), the propeptide was located at the end of the precursor instead of intervening between the signal peptide and mature sequence. SF-im4_1 was identified with 5 S1 and 27 S2 reads. The predicted mature peptide had 25 amino acids with 3 disulfide bonds. It is interesting to note that this superfamily contained the same XXIII (C-C-C-CC-C) framework as the K superfamily [31], but with a much shorter mature sequence. iTRAQ MSMS identified this sequence, although the ratio at peptide level (S1:S2 = 1:2.1) did not correlate well with the cDNA level (S1:S2 = 1:5.4) (Figure 3B).

#### 2.6.18. Superfamily SF-im5

This novel superfamily contained a signal peptide with 54% sequence identity to the O3 superfamily. The total reads for this superfamily were 25 with 4 paralogs identified. SF-im5_1 was identified with 2 S1 and 10 S2 cDNA reads. It contained a classic VI/VII cysteine framework (C-C-CC-C-C), with a predicted cleavage site (RR) that generated mature peptides of 25 amino acids with an amidated C-terminus. iTRAQ MSMS identified SF-im5_1, although the ratio at peptide level (S1:S2 = 1:2.3) did not correlate well with the cDNA level (S1:S2 = 1:5.0) (Figure 3B).

#### 2.6.19. Superfamily SF-im6

This superfamily contained a signal peptide with only 40% sequence identity to the I3 superfamily and was uniquely found in S2, with total reads of 19 and 2 paralogs identified. SF-im6_1 was identified with 0 S1 and 6 S2 reads. The predicted mature peptide of SF-im6_1 had 36 amino acids, 2 disulfide bonds, and a framework of XIV (C-C-C-C). MSMS validated this sequence at the peptide level (Appendix A).

## 3. Discussion

Venom variation occurs widely among venomous animals and appears to be multifactorial in origin [7,10,11,12,13]. Since venom peptides are encoded by small genes and transcribed into mRNA in the venom gland, the relationships between venom variation and toxin mRNA levels may be directly compared within the same animal. The present study utilized a combined transcriptomic and proteomic approach to delineate venom variation and correlate mRNA abundance across two specimens of *C. imperialis*. Transcriptomic analyses revealed 19 conotoxin gene superfamilies, including 17 that were common to both specimens and 6 novel superfamilies (SF-im1 to SF-im6). This is similar to our study of *Conus tulipa* [33], where broad overlap was observed at the gene superfamily level, with two individual specimens having 16 gene superfamilies in common. Consistent with observations for *C. tulipa*, striking individual differences were observed at the sequence level, where a total of 267 conopeptides were identified, with only 62 sequences common to both specimens of *C. imperialis*. These 267 sequences fall within thirteen cysteine-rich frameworks, including 2 new frameworks. Interestingly, of the 46 conopeptide sequences [21,34] described previously in *C. imperialis* venom, only 13 were identified in this study, further highlighting the extent of variation between the venoms of individual cone snails. These “missing” conotoxins included the well-studied ImI [25] and conopressin G [26], suggesting that the toxin expression profile might be linked to many factors, such as geographical variation, as our *Conus imperialis* specimens were collected from pristine reefs in the southern part of the Great Barrier Reef (Australia), in contrast to the previously investigated specimens collected from degraded reefs in the Philippines [25,26].

This is the first time that the predation-evoked venom of a worm-hunting cone snail has been investigated and it shows more complexity (Figure 2A) compared to the simpler predation-evoked venom from piscivorous cone snails, such as *C. striatus* or *C. consors* [15,16,35]. Interestingly, a defense-evoked venom could not be obtained from these specimens, despite repeated attempts. We hypothesize that some worm-hunting cone snails may rely on strong heavy shells for protection and only deploy a defense-evoked venom in response to extreme threats. Alternatively, aquarium conditions may have blunted their defensive response. Consistent with the high level of transcriptomic variability, the toxin profile for the readily obtained predation-evoked milked venom was different between the two animals, confirming observations on extracted venom [36]. However, similar contributions of ~10 major conotoxins from the K and T superfamilies suggest these have been positively selected for prey capture. Despite their potential importance, these toxins have undefined mammalian pharmacology, suggesting they may selectively target worm receptors.

Extending our *Conus tulipa* work, we investigated the relationship between the mRNA levels and peptide levels within the same animal and the levels of translation between the two specimens. ImVC, which was uniquely expressed by S1, was identified only in its corresponding venom duct, but not in the venom of S2 (Table 3). iTRAQ identified four of the ten most abundant conopeptide gene precursors but missed another four, despite their detection in unlabeled venom. Intriguingly, the two most abundant conopeptide gene precursors (A_1 and A_2) remained undetected, possibly due to unusual post-translational modifications that were missed by our current search criteria. However, given that cone snails can inject different venom in defense and predation [17], this suggests these peptides may have a defensive, rather than predatory role. In contrast, the eight other highly transcribed and translated conopeptides showed a strong correlation between the mRNA and predatory venom peptide levels, suggesting these are now subject to stabilizing selection and likely essential for prey capture. However, this correlation failed to extend to peptides transcribed at low and moderate levels, suggesting these conopeptides are not subject to stabilizing selection and alternatively may be subject to diversifying selection. 

In this venomic study, we characterized the *C. imperialis* transciptome and predatory venom proteome. Building on our recent study of variability in *C. tulipa* venom, we examined the extent and nature of individual variation and directly correlated transcription to translation within the same animal. We confirmed that 8 of the 10 most abundant mRNA gene precursors are largely shared by the two specimens and translated to represent the major peptides in the predation-evoked venom, suggesting they are under stabilizing selection for predation. In contrast, transcripts expressed at lower levels showed high levels of variation between specimens and no correlation between transcription and translation within specimens. These results demonstrate that analysis of single specimen provides a unique view of complexity that is lost in pooled samples. The importance of investigating individual venom composition extends beyond cone snails, with a recent venomic study of the venom of rear-fanged snakes demonstrating that pooling of samples masks important individual variation [37]. Presently, the origins of transcriptomic variation and the nature of the various mechanisms involved remain unresolved.

## 4. Materials and Methods

### 4.1. Venom Sample Preparation and RNA Extraction

Two adult specimens of *C. imperialis* collected from the Southern Great Barrier Reef (Queensland, Australia) measuring 6 cm (specimen 1) and 7 cm (specimen 2) were housed in our UQ aquarium facility. The temperature was set to 23–24 °C and a 12:12 light-dark cycle was applied. *C. imperialis* feeds exclusively on a group of polychaete marine worms known as “fireworms” (Amphinomidae) and a milking procedure was adapted from published methods [23,38], employing a live fireworm as a lure and a microcentrifuge tube covered with a piece of the external worm tegument to initiate stinging and collect the predation-evoked venom. Whereas *C. imperialis* has been described previously to release the fired radula tooth immediately upon venom injection [39], we consistently observed that our specimens held onto the radula and reeled back the envenomed worm into their rostrum, reminiscent of hook-and-line fish-hunters (see Appendix A). Furthermore, the quantity of venom injected was remarkable, exceeding previously reported volumes for any cone snail species. Indeed, *C. imperialis* injects up to 180 µL of thick greenish venom, which is different from the translucent venom of hook-and-line fish-hunters or the milky venom of mollusc-hunters. Interestingly, this species was observed to inject venom up to three times within a 30 min trial period. Following the predatory milking procedure, the collected venom was immediately stored at −20 °C. Three days after the milking, the two specimens were dissected on ice. The whole venom duct was quickly removed and placed in a 1.5 mL tube with 1 mL of TRIZOL reagent (Invitrogen). The extraction of total RNA was carried out following the manufacturer’s instructions. The mRNA was purified from the total RNA using a Qiagen mRNA extraction kit. Next generation sequencing using a Roche GS FLX Titanium sequencer was subsequently conducted by AGRF (Australian Genomic Research Facility). The same amount of mRNA (200 ng) was sequenced for both specimens.

Standard Flowgram Format, FASTA, and quality files from 454 containing de-multiplexed read sequences were submitted independently to two different *de novo* assemblers, Newbler v2.6 (Life Science, Frederick, CO, USA) with default parameters (notably, minimum contig length for all contigs = 100, minimum contig length for large contigs = 500, minimum length of reads to use in assembly = 50) and Trinity v2.4.0 [40] in its default configuration and with minimum assembled contig length of 100.

Sorting of cDNA reads and contigs was performed with ConoSorter [20]. Manual validation of conopeptide sequences was then carried out from the retrieved data. Gene superfamilies, signal peptides, and cleavage sites were predicted using the ConoPrec [21] tool, implemented in ConoServer [24]. ConoPrec identifies the signal sequence region using the signal P algorithm [21,41]. The cut-off value for assigning a signal peptide to a gene superfamily was set to >75% sequence identity, as extrapolated from a recent analysis of all precursors deposited in ConoServer [23,24]. Novel superfamilies that showed sequence identity between 50–75% to any known superfamily were considered closely related, whereas no relationship was considered for a sequence identity below 50%.

### 4.2. Proteomic Analysis

Underivatized peptides in the venom samples were analyzed using the SCIEX QStar (Framingham, MA, USA); no concentration-normalization step was undertaken for these TOF scan only samples to ensure a real-time concentration was observed for the milked samples. LC-ESI-MS reconstruction was carried out using Analyst LCMS reconstruct BioTools. The mass range was set between 800 and 12000 Da. Mass tolerance was set as 0.2 Da and S/N threshold was set as 20. The MS level matching of the data was carried out using the ConoMass tools [21]. The precision level was set as 0.5 Da for the automatic matching search. Reconstructed mass lists from the two specimens were compared.

### 4.3. Reduction-Alkylation and Enzyme Digestion

Reduction and alkylation of the cystine bonds was carried out as previously described [28]. Sigma proteomics sequencing grade trypsin and endoproteinase Glu-C were used to digest the reduced and alkylated venom samples and the enzymes were activated in 40 mM NH_4_HCO_3_ buffer. A ratio of 1:100 (w/w) of enzyme to venom peptides was used. The digestion was carried out overnight at 37 °C and enhanced in a microwave apparatus for 4 min on the lowest power setting. 

### 4.4. iTRAQ Labelling, Relative, and Absolute Quantitation of the Injected Venom Samples

The total protein concentration of the two injected venoms was measured using a NanoDrop spectrophotometer (Thermo Fisher Scientific Inc, USA) at 280 nm wavelength. The same calculated amount of total protein (50 µg) was used for each sample (i.e., 2.9 µL of injected venom from specimen 1 and 14.5 µL of injected venom from specimen 2), in order to aid correlation of the protein level of individual conotoxins with the mRNA level and to allow comparison between the two specimens. Reduction, alkylation, and tryptic digestion was carried out as described above, except for the buffer where triethyl ammonium bicarbonate (TEAB) was used instead of ammonium bicarbonate to avoid interference to subsequent labelling. After lyophilization, peptides were labeled with iTRAQ 8-plex reagent (113 for specimen 1 and 115 for the specimen 2) following the manufacturer’s protocol. Individually labeled samples were cleaned up using a cation exchange cartridge, then the two labeled milked venom samples were combined and analyzed using iTRAQ IDA methods [42]. The unlabeled milked venom samples (reduced/alkylated with and without digestion) for each individual were also analyzed by LC-MSMS on the SCIEX 5600 (Framingham, MA, USA). Protein Pilot v4.0.0 software (SCIEX, Framingham, MA, USA) was used for peaklist-generation, sequence identification, and quantitation by searching the LC-ESI-MSMS spectra against a database containing the extracted conopeptides from this work plus published conopeptide sequences (1752 entries). With the alkylated samples, the fixed modification was set as iodoethanol for cysteine alkylation. A total of 9 different types of variable modifications, which have been identified on conopeptides, were considered including the following: Amidation, deamidation, hydroxylation of proline and valine, oxidation of methionine, carboxylation of glutamic acid, cyclization of N-terminal glutamine (pyroglutamate), bromination of tryptophan, and sulfation of tyrosine. The mass tolerance was set as 0.05 Da for precursor ions and 0.1 Da for the fragment ions. Tandem mass spectra were only acquired for the 2 to 5 charged ions. The threshold score for accepting individual peptide spectra was 99. The detected peptide sequences were manually inspected and validated. 

## 5. Conclusions

In this study, we analyzed, for the first time, the predation-evoked venom from a vermivorous species of cone snail and systematically compared two individuals at both transcriptomic and proteomic levels. We demonstrate that transcriptomic sequences expressed at high levels translate to the major peptides in the predatory-evoked venom, while those expressed at low levels correlate poorly with levels of peptide expression. This study contributes to the understanding of the molecular origins of venom variability in cone snails.

## Figures and Tables

**Figure 1 marinedrugs-17-00177-f001:**
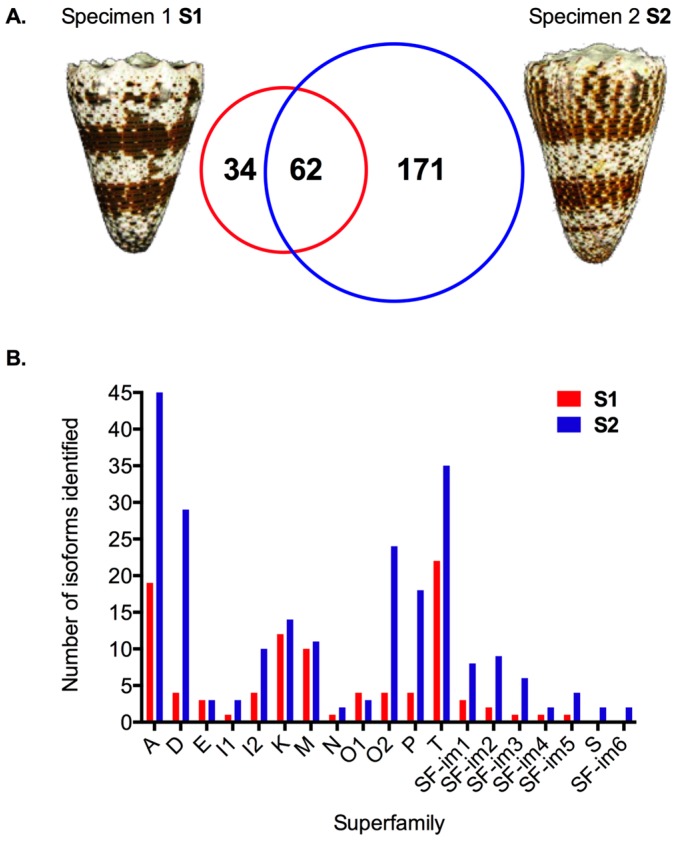
Venom duct transcriptomes of two specimens of *C. imperialis*. (**A**) Variation in the number of transcripts. (**B**) Paralogs of known superfamilies and putative new superfamilies. Color codes: red (Specimen 1, S1) and blue (Specimen 2, S2).

**Figure 2 marinedrugs-17-00177-f002:**
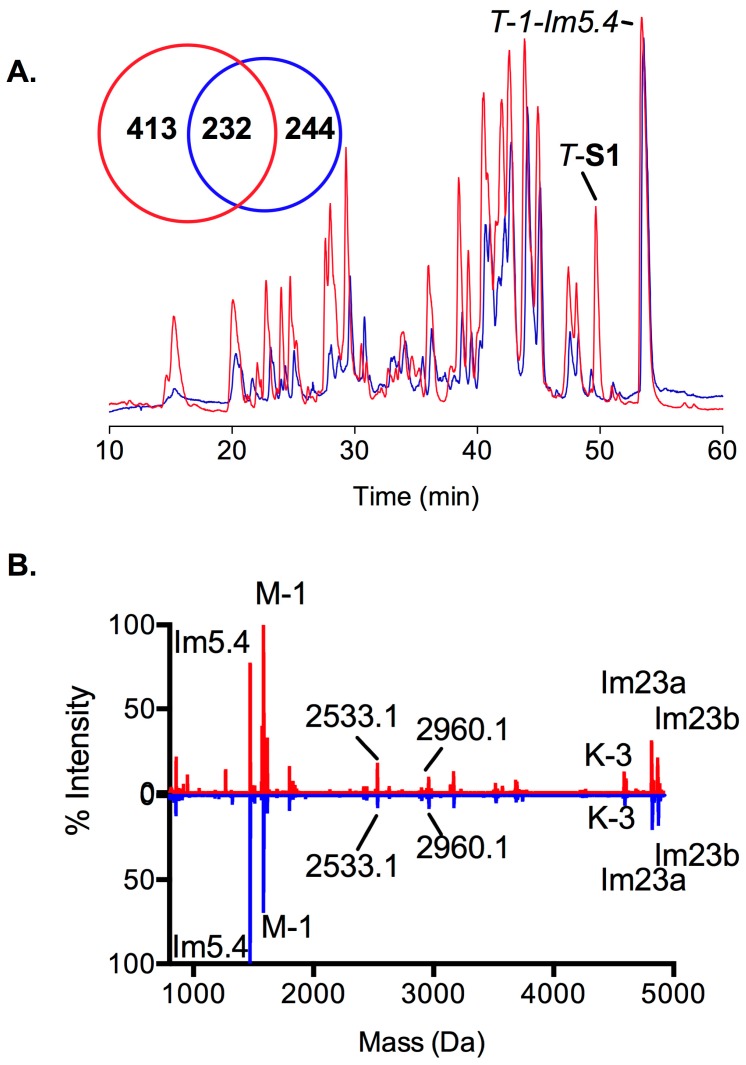
Predation-evoked injected venoms of both *C. imperialis* specimens. (**A**) Overlaid LC-MS profiles of the two specimens. The diagram illustrates the number of masses detected per specimen. *T*-S1 and *T-1-Im5.4* in specimen 1 are labeled on top. (**B**) A normalized figure based on the intensity of the detected masses. Color codes: red (Specimen 1, S1) and blue (Specimen 2, S2).

**Figure 3 marinedrugs-17-00177-f003:**
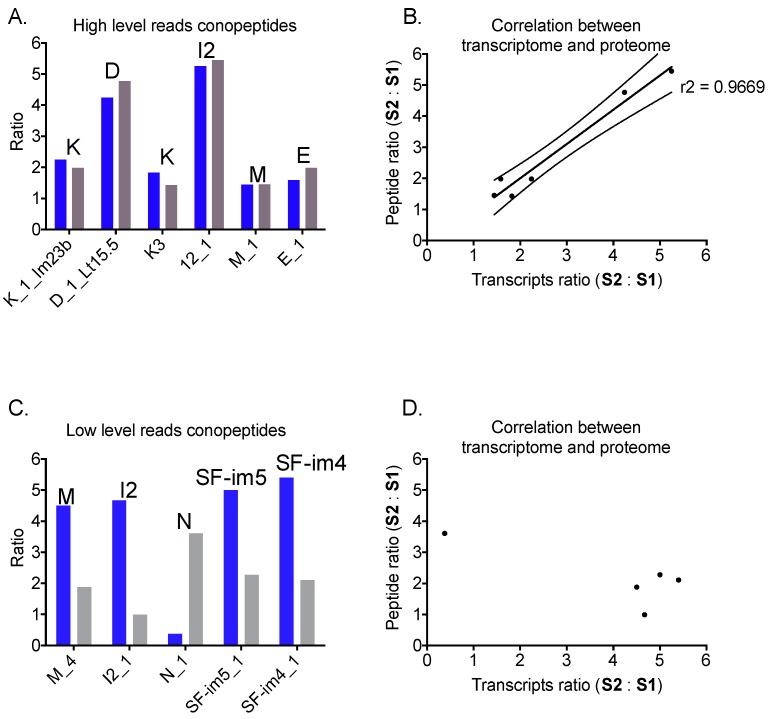
Correlation of mRNA and peptide. (**A**) Abundant conopeptides at mRNA and peptide levels. (**B**) Highly expressed peptides show a good correlation between mRNA level and peptide level. (**C**) Low level conopeptides of mRNAs and peptides. (**D**) Poor correlation for low-level conopeptides. Color codes: blue (ratio of transcripts S2:S1) and black (ratio of peptides S2:S1). Labels on top indicate superfamilies.

**Table 1 marinedrugs-17-00177-t001:** Representative signal sequence and cysteine framework for *C. imperialis* superfamilies.

Gene Superfamily ^1^	Signal Sequence	Cysteine Patterns	Framework
A	MGMRMMFTVFLLVVLATAVLP	CC-C-C	I
D	MPKLEMMLLVLLILPLCYIDA	C-C-C-CC-C-C-C-C-C	Novel
E	MMMRVFIAMFFLLALVEA	C-C-C-C-C-C-C-C	XXII
I1	MKLALTFLLILMILPLMTG	C-C-CC-CC-C-C	XI
I2	MFRVTSVLLVIVLLNLVVLTNA	C-C-CC-CC-C-C	XI
K	MIMRMTLTLFVLVVMTAASASG	C-C-C-CC-C	XXIII
M	MMSTLVVLLTICLLMLPLTA	CC-C-C-CC	III
N	MSTLGMMLLILLLLVPLATFA	C-C-CC-CC-C-C-C-C	Novel
O1	MKLRCMMIVAVLFLTASIFITA	C-C-CC-C-C	VI/VII
O2	MKLTILLLVAALLVLTQA	C-C-CC-C-C	VI/VII
P	MHLSLASSAALMLLLLFALGNFVGVQP	C-C-C-C-C-C	IX
T	MRCLPVVVFLLLLLSAAA	CC-CC	V
SF-im1	MARFLSILLCFAMATGLAAG	C-C	-
SF-im2	MRLTTMHSVILMLLLVFAFDNVDG	C-C-C-C-C-C-CC	Novel
SF-im3	MSKSGMLLFVLLLLLPLAIP	C-CC-C-CC-C-C-C-C	XX
SF-im4	MKFFTCLLLLLVVLTVVFDNVDA	C-C-C-CC-C	XXIII
SF-im5	MKTGMIICLLLIAFMDADG	C-C-CC-C-C	VI/VII
S (**S2**)	MMLKMGAMFAILLLFALSSS	C-C-C-C-C-C-C-C	VIII
SF-im6 (**S2**)	MGVFRCCLAAALVVVCLSRMGG	C-C-C-C	XIV

^1^ The top 17 superfamilies were found in both specimens and the bottom 2 superfamilies (S and putative SF-im6) were identified only in specimen 2, S2.

**Table 2 marinedrugs-17-00177-t002:** The total number of paralogs and cDNA reads for each superfamily in individual specimen.

Gene Superfamily	Total Reads	Total Paralogs	Common Paralogs
A	5826	52	15
D	2421	29	4
E	204	3	3
I1	55	3	1
I2	1426	10	4
K	1032	22	6
M	498	17	4
N	34	2	1
O1	76	6	1
O2	1038	25	3
P	400	18	4
S	19	2	0
T	2166	46	11
SF-im1	85	10	1
SF-im2	81	10	1
SF-im3	73	6	1
SF-im4	37	2	1
SF-im5	25	4	1
SF-im6	11	2	0

**Table 3 marinedrugs-17-00177-t003:** Correlation of two conopeptides between mRNA and peptide levels within the same animal (specimen 1, S1).

Name	Mature Peptide Sequence	cDNA Reads		MSMS Precursor Intensity	
		Full Precursor	Mature Peptide	RM Only	RM and Digested
*T*_S1 *(ImVC)*	**FLNTICCWSR^1^ACCG-NH_2_**	41	137	1839217	22,290
*T_1_Im5.4*	**FLNTICCWSGACCG-NH_2_**	491	1297	7,744,866	1,248,633
*Ratio*		12	9	4	56

^1^ Arginine in position 10 is in red.

**Table 4 marinedrugs-17-00177-t004:** Representative conotoxin precursors, mature peptides and cysteine frameworks. The predicted mature peptides are in bold. Cysteine frameworks are highlighted in orange. Sequences have been published in NCBI database (GenBank KT377395-KT377426).

Name	Seq No.	Precursor Sequence
A_1	im001	MGMRMMFTVFLLVVLATTVVPITLASATDGRNAAADARMSPLISKFKK-**DYCHKYGYTI**G
A_2	im002	MGMRMMFTVFLLVVLATTVVPITLASATDGRNAAANARVSPVISKFSKK-**WCHPNPYTV**G
A_3_Bn1.3	im003	MGMRMMFTVFLLVVLATAVLPVTLDRASDGRNAAANAKTPRLIAPFIR**DYCCHRGPCMVWC**G
D_1_Lt15.5	im004	MPKLEMMLLVLLILPLCYIDAVGPPPPWNMEDEIIEHWQK**LHCHEISDLTPWILCSPEPLCGGKGCCAQEVCDCSGPVCTCPPCL**
E_1	im005	MMMRVFIAMFFLLALVEAGWPRLYDK**NCKKNILRTYCSNKICGEATKNTNGELQCTMYCRCANGCFRGQYIDWPNQQTNLLFC**
I1_1	im006	MKLALTFLLILMILPLMTGEKTSDDLELRGVESLRAIFRDRR**CSDNIGATCSDRFDCCGSMCCIGGQCVVTFAECS**
I2_1	im007	MFRVTSV--LLVIV-LLNLVVLTNA**CHMD---CSKMT-CCSGICCFY-CGRPMCPGT**RRALLQRLVGHQR
I2_2	im008	MFRLTSVGCILLVIAFLNLVGLTNA**CTSEGYSCSSDSNCCKNVCCWNVCESH-CRHP**GKRTRLQGFFKHRR
K_1_Im23b	im009	MIMRMTLTLFVLVVMTAASASGDALTEAKR**IPYCGQTGAECYSWCIKQDLSKDWCCDFVKTIARLPPAHICSQ**
K_2_Im23a	im010	MIMRMTLTLFVLVVMTAASASGDALTEAKR**IPYCGQTGAECYSWCIKQDLSKDWCCDFVKDIRMNPPADKCP**
K_3	im011	MIMRMTLTLFVLVVMTAASASGDALTEAKR**VPYCGQTGAECYSWCKEQHLIR--CCDFVKYVGMNPPADKCR**
M_1	im012	MMSTLVVLLTICLLMLPLTARQLDADQLADQLAERMEDISADQNRWFDPVKR**CCMRPICM-----CP-CCVN**G
M_2_Eu3.3	im013	MMSKLGVLLAICLLMLPLTALPLDGDQPQER---KEDGKSAALQPWFDPVKR**CCQAA-CSPWL--CLPCC**G
M_3_Bt3.1	im014	MMSTLGVLLTIGLLLFPLTALPLDGDQPADQPAERLQDISPKEIPGSDPFKR**CCHAPYCTPPHLGC-PCC**GK
M_4	im015	MSKVGVVPLIFLVLLSIAALQNGDDPRRQRDEKQSPQGDILRSTLTKYSYNIQRR**CWAGGSPCHLCSSSQVCIAPTGHPAIMCGRCVPILT**
N_1	im016	MSTLGMMLLILLLLVPLATFADDGPTMRGHRSAKLLAHTTR**DSCPSGTNCPSKICCNGNCCSKSSCRCETNQATKERVCVC**
O1_1_Conotoxin3	im017	MKLRCMMIVAVLFLTASIFITADNSRNGIENLPRMRRHEMKKPKASKLNKR**GCLPDEYFCGFSMIGALLCCSGWCLGICMT**
O2_1_im6.2	im018	MKLTILLLVAALLVLTQARTERRRVKSRKTSSTYDDEMATFCWSY**WNEFQYSYPYTYVQPCLTLGKACTTNSDCCSKYCNTKMCKINWEG**
O2_3	im019	MEKLTILLLVTAVLMSTQALMQSGIEKRQRAKIKFFSKRKTTAER----------**WWEGECYDWLRQCSSPAQCCSGNCGAH-CKAW**
P_1	im020	MHLSLASSAALMLLLLFALGNFVGVQPGQIRDLNKGQLKDNRRNLQSQRK**QMSLLKSLHDRNG-CNGNTCSNSP-CPNNCY-CDTEDD---CHPD**RREH
P_2	im021	MHLSTASSVALMFFLLFAFYGVQPELMTRDVDNGQLTDNRRNLRSRVKPTGLFKSRK--**PSED-C-GKTCETAENCPDDCSSCLSVEGTYRCA**
P_3	im022	MHRSLAGSAVLMLLLLFALGNFVGVQPGLVTRDADNGQLMDNRRNLRLERKTMSLFKSLDKR**ADCSTY-CFGMGICQSGCY-CGPGHA---CMPN**GR
T_1_Im5.4	im023	MRCLPVVVFLLLLLSAAAAPGVGSKTER**LPGLTSSGDSDESLP---FLNTICCWSGA-CCG**G
T_2	im024	MCCIPVFFILLLLIPSAPSILAQPTTKGDVALASSYDDAKR**TLQRLSIKYSCCPGIVSCCVIP**
SF-im1_1	im025	MARFLSILLCFAMATGLAAGIRYPDRVLGRCSTHDLSKMEIDTNLDGVYSPHRSFCTCGSGEVYFTAKDRR**NHSNYRVYVCGMPTEFCTAENPVRDP**
SF-im2_1	im026	MRLTTMHSVILMLLLVFAFDNVDGDEPGQTARDVDNRNFMSILRSEGKPVHFLRAIKKR**DCTGQACTTGDNCPSECVCNEHHFCTGKCCYFLHA**
SF-im3_1	im027	MSKSGMLLFVLLLLLPLAIPELAPAGRSVTHHFRDFGAKR**SVPISCVNPSTPNLQGSWQDKKCCSTKLCSPTNCCESSTCSCVEGSCQCL**
SF-im4_1	im028	MKFFTCLLLLLVVLTVVFDNVDA**CDRSCTGVMGHPSCATCCACFTSAG**KRHADGQHSRMKVRTGAKNLLKRMPLH
SF-im5_1	im029	MKTGMIICLLLIAFMDADGSPGDTLYSQKTADTDSGMKRFQKTFQKRR**CVFCPKEPCCDGDQCMTAPGTGPFC**G
S_1	im030	MMLKMGAMFAILLLFALSSSQQEGDVQARKIRLRNDFLRTSRMIFTR**GCGGSCHTSPGCGGNCECNSPVPCYCSGTETCVCVCS**G
SF-im6_1	im031	MGVFRCCLAAALVVVCLSRMGGTEPLESNHEDERR**ADDTSGDDCVDTNEDCVNWASTGQCEANPSYMRENC**RK

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
