# Peer review of "Transcriptomic-Proteomic Correlation in the Predation-Evoked Venom of the Cone Snail, Conus imperialis"

_marinedrugs, 2019, doi:10.3390/md17030177_

Round 1

Reviewer 1 Report

The paper entitled “Transcriptomic-proteomic correlation in the predation-evoked venom of the cone snail, Conus imperialis” by Jin et al. presents a survey of the mRNA precursor transcripts of the venom glands and the conotoxin peptides of the milked venoms of two individuals of Conus imperialis.  While it has been always assumed that there should be a correlation between diversity of both products, this is the first study that investigates such hypothesis.  Up to 267 precursors were identified belonging to 17 known and 5 new superfamilies. The main conclusion of the paper is that there is strong correlation between levels of transcription and translation for highly expressed (in terms of number of reads) transcripts but not for low expressed mRNAs.

The paper is well written, the methodology is appropriate (but see below) and conclusions are supported by data and analyses. The new reported conotoxin precursors of Conus imperialis are of interest on their own and for comparison with venom cocktails from other species, and the demonstration of a strong correlation between the transcriptome and proteome for most abundant cDNAS sets the standards for other studies on cone venomics. For all these reasons, I feel that the study should be published in Marine drugs and will be of the interest to its wider readership.

I have the following suggestions that the authors may want to take into consideration when revising the manuscript:

-       Line 23. Perhaps iTRAQ is too technical for an abstract. Could the authors substitute the word by e.g. labeled peptides or similar?

-       Line 53. Could the authors introduce here that Conus imperialis are vermivorous and in particular they eat amphinomids (as shown much later in the M&M)?

-       Line 72. Please refer here to the total number of different conotoxins found in Conus imperalis i.e., 267 (as shown later in the discussion)

-       Line 66. Inconsistent? Could you give more details?  Now the 454 technology is outdated and newly reported transcriptomes are based on the assembly of much shorter reads from Illumina, so some discussion (in the corresponding section) on this issue would be interesting.

-       Line 74. Homology is a statement of similarity due to shared ancestry. Hence, it cannot be measured in percentage. Here, and in other instances of the manuscript (e.g., lines 87 and 88, 253, 260, 265, 272, 281, 287), the authors are in fact dealing with percentage of similarity or identity. Please correct.

-       Line 81. Level of transcription could also refer to expression data and not diversity data as implied here. Perhaps it would be more precise saying variation in the number of transcripts.

-       Lines 86- 88. Which is the threshold to declare that a new superfamily is  not closely related (below 50%?) or related (between 50-75%?) to any known SF? For those cases in which the first hit is 50-75%, which is the value of the second hit? Is there a significant difference to assign a potential relationship to a known SF?

-       Line 94. The last two columns of the table indicate the same numbers in Fig 1b so they could be redundant. In any case, e.g., SFA in S2 has 45 isoforms in the figure and 48 in the table. Please double check.

-       Lines 116-119. This is discussion rather than results.

-       Lines 129-130 and 131-132. These two sentences could be merged into a single one.

-       Line 163. What do the hyphens in the sequences mean? Could you add a column with the framework to the table?

-       Line 179.  For low expressed transcripts (Fig. 3 B ii). It looks that there is a correlation but negative. Have you tried to estimate the regression line and the r2?

-       Line 180. In Fig 3, the authors show high and low expressed transcripts and corresponding peptides. Could you list altogether somewhere those precursors that are later not found in the proteome at all?

-       -Line 226. “ancestral” here is not a proper evolutionary wording. The O1 superfamily is found in early diverging taxa within Conidae such as Conus californicus (and can be assumed to be widespread in cones) but the exact composition of the venom cocktail of the ancestor can only be hypothesized.

-       Lines 275-278. This is discussion rather than results.

-       Line 298. Could you discuss here why one specimen had almost double precursors than the other (see lines 96-98 and line 123)? Is it natural variation or could there have been any bias during the processing of the samples (having different number of total reads)?

-       Line 333. And are essential for prey capture?

-       Lines 363 and 400.  A Greek character is missing, perhaps micro (µ)?

-       Line 440. I miss reference to and some discussion of the paper “Diversity and evolution of conotoxins in Conus virgo, Conus eburneus, Conus imperialis and Conus marmoreus from the South China Sea” by Liu et al (2012) Toxicon 60: 982 that also reports transcript from C. imperialis.  

Author Response

Reviewer 1

The paper entitled “Transcriptomic-proteomic correlation in the predation-evoked venom of the cone snail, Conus imperialis” by Jin et al. presents a survey of the mRNA precursor transcripts of the venom glands and the conotoxin peptides of the milked venoms of two individuals of Conus imperialis.  While it has been always assumed that there should be a correlation between diversity of both products, this is the first study that investigates such hypothesis.  Up to 267 precursors were identified belonging to 17 known and 5 new superfamilies. The main conclusion of the paper is that there is strong correlation between levels of transcription and translation for highly expressed (in terms of number of reads) transcripts but not for low expressed mRNAs.

The paper is well written, the methodology is appropriate (but see below) and conclusions are supported by data and analyses. The new reported conotoxin precursors of Conus imperialis are of interest on their own and for comparison with venom cocktails from other species, and the demonstration of a strong correlation between the transcriptome and proteome for most abundant cDNAS sets the standards for other studies on cone venomics. For all these reasons, I feel that the study should be published in Marine drugs and will be of the interest to its wider readership.

I have the following suggestions that the authors may want to take into consideration when revising the manuscript:

-       Line 23. Perhaps iTRAQ is too technical for an abstract. Could the authors substitute the word by e.g. labeled peptides or similar?

Following the reviewer’s suggestion, we changed iTRAQ to "quantitative labeling method".

-       Line 53. Could the authors introduce here that Conus imperialis are vermivorous and in particular they eat amphinomids (as shown much later in the M&M)?

We agree and now have added line 54-57, the following sentence: "C. imperialis is a worm-hunting species that feeds exclusively on a group of polychaete marine worms known as “fireworms” (Amphinomidae)."

-       Line 72. Please refer here to the total number of different conotoxins found in Conus imperalis i.e., 267 (as shown later in the discussion)

As suggested, the sentence line 76-78 now reads: "a total of 267 unique conopeptide precursors were retrieved, including 96 from the specimen S1 and 233 from specimen S2"

-       Line 66. Inconsistent? Could you give more details?  Now the 454 technology is outdated and newly reported transcriptomes are based on the assembly of much shorter reads from Illumina, so some discussion (in the corresponding section) on this issue would be interesting.

The main reason we use reads rather than contigs is that the short sequences of conotoxin precursors (70-100 a.a. on average) are fully covered by most 454 reads (average length 390 bp) (Dutertre et al., 2013) and thus we have the benefit of being able to use only these full length reads.. Nonetheless, to address the reviewer’s concern, the raw data have been assembled using two different softwares, namely Newbler 2.5 and Trinity. In the Supplementary Table 3 we compared the results from the two assemblers (Trinity and Newbler), together with the raw data and our proteomics data.

The following discussion has been added after Supplementary Table 3.

"The major conclusions drawn from the assembly work are:

1) The average length after assembly was not much different compared to the raw reads, e.g. the trimmed raw reads average length was ~390 bp, after the Trinity assembly it is ~480 bp, after Newbler assembly, it is 430-460 bp. The average length for our conopeptide precursors is ~300 bp, confirming that analyzing the raw 454 reads would be sufficient to cover the full length of our conopeptides.

2) From the 62 common sequences retrieved initially from the raw 454 read datasets of both specimens, only 15 (Newbler) or 18 (Trinity) could be found after the assembly, and different assembler produce different sequences!

3) Whereas the majority of the low expressed sequences have disappeared after the assembly as expected (majority of the red zone), it was quite surprising that some highly expressed sequences could not be found anymore, e.g. the most abundant sequence T_1_Im5.4 (1027 reads) went missing from the specimen S2 after Newbler assembly, and the major T-S1 (the T peptide only found in specimen S1) also disappeared after both Trinity and Newbler assembly.

4) More importantly, some of the missing sequences after assembly had been unambiguously confirmed by MSMS, highlighting a major issue with current assembly algorithms, as this approach can eliminate valid sequences."

While increasing the depth of the analysis using Illumina can collapse sequences with errors into their parental types to improve the quantitative assessment of clustering and sequence assembly this approach is beyond the scope of the present study.

-       Line 74. Homology is a statement of similarity due to shared ancestry. Hence, it cannot be measured in percentage. Here, and in other instances of the manuscript (e.g., lines 87 and 88, 253, 260, 265, 272, 281, 287), the authors are in fact dealing with percentage of similarity or identity. Please correct.

As suggested, it was changed to "sequence identity" thoroughly.

-       Line 81. Level of transcription could also refer to expression data and not diversity data as implied here. Perhaps it would be more precise saying variation in the number of transcripts.

Changed to "variation in the number of transcripts".

-       Lines 86- 88. Which is the threshold to declare that a new superfamily is  not closely related (below 50%?) or related (between 50-75%?) to any known SF? For those cases in which the first hit is 50-75%, which is the value of the second hit? Is there a significant difference to assign a potential relationship to a known SF?

Indeed, we considered a novel superfamily not closely related to any other superfamily if the percentage of sequence identity was below 50%, and closely related between 50-75%. This is now indicated in the experimental section line 416-417. The close relationship between superfamily may suggest a common origin, but the significance of the relationship is unclear in most cases (even less so for the second hit).

-       Line 94. The last two columns of the table indicate the same numbers in Fig 1b so they could be redundant. In any case, e.g., SFA in S2 has 45 isoforms in the figure and 48 in the table. Please double check.

Thank you for pointing this out, we have now corrected Figure 1b and deleted the last two columns of Table 2.

-       Lines 116-119. This is discussion rather than results.

The reviewer refers to our sentence “it is the first time that the predation-evoked venom of a worm-hunting cone snail has been investigated, and it shows more complexity (Figure 2A) compared to the simpler predation-evoked venom from piscivorous cone snails such as C. striatus or C. consors (Jakubowski et al., 2005, Biass et al., 2009, Dutertre et al., 2010). We agree and have moved the sentence to the discussion line 343-345.

-       Lines 129-130 and 131-132. These two sentences could be merged into a single one.

As suggested the two sentences have been merged: Indeed, the precursor of ImVC (T-S1), the second most highly expressed superfamily T peptide (1570.66 Da; FLNTICCWSRACCG-NH2), was found in the venom gland transcriptome of S1 but not in S2, and this peptide was relatively abundant in the predation-evoked venom of S1 but remarkably absent from the corresponding venom of S2.

-       Line 163. What do the hyphens in the sequences mean? Could you add a column with the framework to the table?

The sequences shown in table were retrieved from superfamily alignments, hence the hyphens were gaps within the alignments that were carried on to the table.

-       Line 179.  For low expressed transcripts (Fig. 3 B ii). It looks that there is a correlation but negative. Have you tried to estimate the regression line and the r2?

Yes, we have attempted to estimate the regression line but there is none for the low expressed transcripts.

      Line 180. In Fig 3, the authors show high and low expressed transcripts and corresponding peptides. Could you list altogether somewhere those precursors that are later not found in the proteome at all?

Precursors validated by MS/MS are shown in supplementary table 2. Therefore, all other precursors not in this table were not validated at the proteomic level.

-       -Line 226. “ancestral” here is not a proper evolutionary wording. The O1 superfamily is found in early diverging taxa within Conidae such as Conus californicus (and can be assumed to be widespread in cones) but the exact composition of the venom cocktail of the ancestor can only be hypothesized.

As suggested, the sentence now reads: Surprisingly, though O1 is one of the most widespread superfamily in cones and found in early diverging taxa such as C. californicus…”

-       Lines 275-278. This is discussion rather than results.

In line 304-305, the first sentence is descriptive, whereas the second one, we agree reflect our interpretation and has been removed. “It is interesting to note that this superfamily contained the same XXIII (C-C-C-CC-C) framework as the K superfamily[33], but with a much shorter mature sequence. It would be interesting to see if the double helical structure for framework XXIII (Im23a and Im23b) is still maintained for the mature peptide of SF-im4_1.”

-       Line 298. Could you discuss here why one specimen had almost double precursors than the other (see lines 96-98 and line 123)? Is it natural variation or could there have been any bias during the processing of the samples (having different number of total reads)?

Fragment analyzer results on both RNA extractions show similar quality, therefore we believe this is natural variation and not due to experimental bias. In addition, we have found similar results in our recently published study on C. tulipa venom(Dutt et al., 2019) and others have also reported significant variations between specimens in other species(Abalde et al., 2018).

-       Line 333. And are essential for prey capture?

We have modified the sentence of lines 366-367 according to the reviewer’s suggestion: “…suggesting these are now subject to stabilising selection and likely essential for prey capture.”

-       Lines 363 and 400.  A Greek character is missing, perhaps micro (µ)?

Thank you, it seems indeed that the pdf conversion omitted the “micro” symbols. We have changed all back to the correct symbols.

-       Line 440. I miss reference to and some discussion of the paper “Diversity and evolution of conotoxins in Conus virgo, Conus eburneus, Conus imperialis and Conus marmoreus from the South China Sea” by Liu et al (2012) Toxicon 60: 982 that also reports transcript from C. imperialis.  

Thank you, the reference has been added.

Reviewer 2 Report

Authors analyzed two different specimens of the same species of Conus, using proteomics and transcriptomics in order to detect differences between the transcribed DNA and the mature proteins.

Authors of the same group, had been previously established that depending on the part of the venom duct that you select, you will find different proteins. However, they don´t mention if they isolate the RNA from the whole duct, or from one part of the venom duct, please clarified this point.

Author Response

Reviewer 2

Authors analyzed two different specimens of the same species of Conus, using proteomics and transcriptomics in order to detect differences between the transcribed DNA and the mature proteins.

Authors of the same group, had been previously established that depending on the part of the venom duct that you select, you will find different proteins. However, they don´t mention if they isolate the RNA from the whole duct, or from one part of the venom duct, please clarified this point.

The RNA was extracted from the whole venom duct, this is now specifically indicated in the Materials and Method section.

Reviewer 3 Report

Dear Authors,

Present in the manuscript is a large high-quality proteomic and transcriptomic dataset for this conus specie. The manuscript is written well and data is presented in a systematic manner.

 Shown are some very interesting corelations between the transcriptomic and proteomic data between the low reads for two different individuals of  the same specie. 

The study will help in understanding  intraspecie venom variability. 

Few questions can be answered:

I understand the title says "Transcriptomic-proteomic correlation in the predation-evoked venom of the cone snail" but

1. Did the authors try to compare the novel super families which were found in this species with turripeptides or terepeptides super familes as they are known to have super families distinct from the cone snail super families.

2. Similarly, the novel cysteine frameworks were compared with the frameworks outside the cone snail cys frameworks or not?

3. Looks like there is typographical errors here and there. 

For eg: Line 399 and 400: the units need to be corrected, I can't think of authors having Liters of venom and grams of proteins.

Best of luck

Author Response

Reviewer 3

Present in the manuscript is a large high-quality proteomic and transcriptomic dataset for this conus specie. The manuscript is written well and data is presented in a systematic manner.

Shown are some very interesting corelations between the transcriptomic and proteomic data between the low reads for two different individuals of  the same specie. 

The study will help in understanding  intraspecie venom variability. 

Few questions can be answered:

I understand the title says "Transcriptomic-proteomic correlation in the predation-evoked venom of the cone snail" but

1. Did the authors try to compare the novel super families which were found in this species with turripeptides or terepeptides super familes as they are known to have super families distinct from the cone snail super families.

We acknowledge the reviewer’s comment, Teretoxins have many similar features to conopeptides such as the organization of the teretoxin transcripts also contain signal peptide, propetide and mature peptide region. With approximated 300 different species of terebrids and multiple new superfamilies of teretoxins identified from just a few species (Turner et al., 2018, Gorson et al., 2015), we feel that such comparison to our superfamilies is beyond the scope of our study.

2. Similarly, the novel cysteine frameworks were compared with the frameworks outside the cone snail cys frameworks or not?

A high diversity of cysteine frameworks have been discovered from teretoxins, similar to the reason listed above, we feel that such comparison to our frameworks is beyond the scope of our study.

3. Looks like there is typographical errors here and there. 

For eg: Line 399 and 400: the units need to be corrected, I can't think of authors having Liters of venom and grams of proteins.

The pdf conversion omitted the “micro” symbols. The manuscript has been thoroughly checked for typos.

Reviewer 4 Report

General comments to authors:

The authors present an interesting study combining transcriptomics and proteomics to characterize the venom gland transcriptome and predatory venom proteome of Conus imperialis in order to evaluate whether there is variability within and between individuals. This is also the first time that the predation-evoked venom of a vermivorous cone snail is investigated, showing that it is more complex than the venom of piscivorous cone snails. The results are interesting, suggesting that there is correlation between highly expressed toxin precursor transcripts and major peptides in the predatory-evoked venom, while transcripts expressed at low levels correlate poorly with peptide expression levels in the venom. This is an interesting finding, suggesting a possible mechanism contributing to individual venom variation, namely stabilizing selection acting on a handful of toxins that might represent key venom components, whereas diversifying selection might promote gene transcription divergence and variation among less expressed venom components. I think this is a very valuable study, providing new and interesting data about an important question in venomics research, namely what the drivers of venom variation are. However, I have some concerns and comments that I would like the authors to address.

My main concern is regarding the methods to estimate transcript abundance. It is not explained in the Methods section how are mRNA levels being estimated for each conotoxin precursor. It seems that the authors are using total number of cDNA reads, but this is rather confusing since they did not assemble a venom gland transcriptome, but instead sorted the raw reads using ConoSorter. In order to use read counts as a proxy for mRNA abundance, one typically builds a transcriptome and then maps the raw reads back to this transcriptome to calculate measurements such as RPKM or FPKM which consider the length of each transcript (as longer transcripts will have more reads mapped just because they are longer). Authors should include an additional section in the Methods describing their transcriptomic analysis in more detail and including missing information to allow a proper evaluation of their methods. If it has not been done, I suggest that authors assemble the venom gland transcriptomes and estimate transcript abundance by mapping reads using RSEM or an equivalent program that takes into account trasncript length to normalize measurements and allow for appropriate comparisons.

Specific comments:

-       Comment 1: Authors state in the Results section (P2, L70-71) that it is difficult to distinguish biological variation from sequencing errors when coverage is low, yet it is not clear whether the employed sequencing depth is enough to reduce this issue. I would suggest that the authors include this information in order to better evaluate the suitability of their approach. I was also wondering why the authors chose 454 technology as opposed to Illumina, since it has been demonstrated that the latter achieves greater depth of coverage, increased variant sensitivity and fewer false positives. If there is a specific reason for which 454 performs better in this particular case, the authors should clearly specify that in the manuscript.

-       Comment 2: Authors make use of the mistaken concept “percentage homology” throughout the text (P2, L74; P4, L.87-88). Homology is not a measurable feature, but a quality that indicates shared ancestry. Thus, two sequences are either homologous or are not, but cannot be 50% homologous. This term should be changed to “sequence identity” throughout the manuscript.

-       Comment 3: The terms paralog and isoform are used interchangeably throughout the manuscript, yet they are not equivalent (for example P4, L92; Table 2). Authors should clearly state whether they are reporting paralogs or isoforms, and how are they defining each. There is already variation in the literature when referring to protein isoforms that leads to confusion, and thus authors should clarify this point.

-       Comment 4: In the section Transcriptomic intraspecific variation, authors state that they found more isoforms and recovered 1.5 times as many cDNA reads in S2 than in S1 (P4, L96-98). This could be the result of lower integrity and quality of the isolated mRNA and the resulting cDNA libraries from one of the samples, which would also require a more stringent trimming and removal of a greater number of raw reads. Authors should also report the initial number of raw reads per sample and the RNA RIN or other appropriate measurement of RNA quality/integrity measurement.

-       Comment 5: In the Discussion authors argue (P11, L308) that certain conotoxins previously found in C. imperialis but missing in their study might not be critical components of the venom. I think authors should revise this statement as there are many factors that could account for this discrepancy, for example, these particular conotoxins might be important components in the defensive venom and not in the predatory venom, or as they state in the very next sentence, the differences might also be due to geographical variation, or even size or life stage of the analyzed specimen.

Other minor comments:

-       P1, L36 – Missing period after [14]

-       P2, L53 – This is the first time that C. imperialis is mentioned in the text and therefore the genus name should be written in full.

-       The μ sign is missing in several places throughout the manuscript (P12, L363; P13, L399-400).

Author Response

Reviewer 4:

The authors present an interesting study combining transcriptomics and proteomics to characterize the venom gland transcriptome and predatory venom proteome of Conus imperialis in order to evaluate whether there is variability within and between individuals. This is also the first time that the predation-evoked venom of a vermivorous cone snail is investigated, showing that it is more complex than the venom of piscivorous cone snails. The results are interesting, suggesting that there is correlation between highly expressed toxin precursor transcripts and major peptides in the predatory-evoked venom, while transcripts expressed at low levels correlate poorly with peptide expression levels in the venom. This is an interesting finding, suggesting a possible mechanism contributing to individual venom variation, namely stabilizing selection acting on a handful of toxins that might represent key venom components, whereas diversifying selection might promote gene transcription divergence and variation among less expressed venom components. I think this is a very valuable study, providing new and interesting data about an important question in venomics research, namely what the drivers of venom variation are.

However, I have some concerns and comments that I would like the authors to address.

My main concern is regarding the methods to estimate transcript abundance. It is not explained in the Methods section how are mRNA levels being estimated for each conotoxin precursor. It seems that the authors are using total number of cDNA reads, but this is rather confusing since they did not assemble a venom gland transcriptome, but instead sorted the raw reads using ConoSorter.

We agree that the lack of precision when using cDNA read counts as a measure of mRNA expression can introduce additional imprecision. However, we report here the abundance of identical trimmed cDNA reads showing the structure (i.e. presence of signal, pro- and mature regions) of precursor conotoxins and did not use the raw reads directly, which could have indeed introduced additional imprecision. Using precursor conotoxin sequences measured at the cDNA level allows a clear distinction between high and low levels of expressed transcripts. In addition, we also assembled the raw reads and explained the problem of the assembled data in answer to reviewer 1 which was added to the supplementary discussion.

 In order to use read counts as a proxy for mRNA abundance, one typically builds a transcriptome and then maps the raw reads back to this transcriptome to calculate measurements such as RPKM or FPKM which consider the length of each transcript (as longer transcripts will have more reads mapped just because they are longer). Authors should include an additional section in the Methods describing their transcriptomic analysis in more detail and including missing information to allow a proper evaluation of their methods. If it has not been done, I suggest that authors assemble the venom gland transcriptomes and estimate transcript abundance by mapping reads using RSEM or an equivalent program that takes into account transcript length to normalize measurements and allow for appropriate comparisons.

During mRNA library prep, one usually use several runs of end-point PCR amplification (as opposed to real-time or quantitative PCR that allows to measure the initial quantity of RNA). Depending on the primary structures of the RNAs and the primers, the efficiency of the amplification is different from one transcript to another. Thus, the amount of amplified RNA is not proportional to the number of cycles or incubation time. Moreover, PCR also generates transcript duplicates that incorporates a bias when estimating RNA expression (along with optical duplicates produced by the sequencer). Consequently, following the library prep, the original mRNA quantities can be distorted. 

Assessing mRNA expression could be done according to the following steps:

(1) Trimming the raw reads by quality

(2a) Aligning the reads using a splice-aware aligner (e.g. STAR) to a reference genome or transcriptome generated from the same species but unfortunately these datasets are not yet available for cone snails, let alone C. imperialis.

OR

(2b) Performing a de novo assembly of the transcriptome from the sequencing reads. In practice, none of the de novo assemblers we tested in our PNAS paper (Lavergne et al., 2015) or the ones we used for the purpose of this paper (see details above) led to confident results. Even though such a performant assembler would exist, one could not reconstruct the cone snail transcriptome using a poly-A tail RNA selection strategy, since the isolated mature transcripts could have been alternatively spliced (assuming such a RNA maturation mechanism exists in cone snail, which still need to be confirmed). Therefore, full transcriptome sequencing with high sequencing depth would be necessary to cover as much as possible the entire transcriptome or genome.

(3) Annotating the aligned transcripts (i.e. put a gene name on each reads). Once a genome or transcriptome has been generated, gene annotations would still need to be produced (to know which transcript corresponds to which functional location of the genome). However, focusing on conotoxins identification only, the use of ConoSorter algorithm could overpass this limitation.

(4) Removing PCR and optical duplicates (although it is arguable for transcriptomes since the distinction between technical replicates and gene expression would be unclear…).

(5) Counting the conotoxin reads (HTSeq) or estimating their number (RSEM, or more recent pseudo-count algorithms such as Kallisto or Salmon). At this step, an expression matrix (i.e. a table where rows are genes / columns are samples) can be built.

(6) Normalizing the read count matrix by sequencing depth and gene length in order to compare the RNA expression within and between samples (for gene length, once again, a reference genome and proper gene annotations would be needed). Different normalization methods exist for removing batch effects, depending on the purpose of the study (e.g. as Reviewer mentioned, expression can be measured in FPKM - fragment per kilobase per million, for paired-end, also called RPKM for single-end reads, a unit that has been contested over the past few years; TPM – transcript per million - is now a preferred unit(Wagner et al.)

While we value reviewer's suggestion, we feel this approach is beyond the scope of the present study.

Specific comments:

-       Comment 1: Authors state in the Results section (P2, L70-71) that it is difficult to distinguish biological variation from sequencing errors when coverage is low, yet it is not clear whether the employed sequencing depth is enough to reduce this issue. I would suggest that the authors include this information in order to better evaluate the suitability of their approach. I was also wondering why the authors chose 454 technology as opposed to Illumina, since it has been demonstrated that the latter achieves greater depth of coverage, increased variant sensitivity and fewer false positives. If there is a specific reason for which 454 performs better in this particular case, the authors should clearly specify that in the manuscript.

The main reason we have been using reads rather than contigs is that the short sequences of conotoxin precursors (70-100 a.a. on average) are fully covered by most 454 reads (average length 390 bp) (Dutertre et al., 2013).

-       Comment 2: Authors make use of the mistaken concept “percentage homology” throughout the text (P2, L74; P4, L.87-88). Homology is not a measurable feature, but a quality that indicates shared ancestry. Thus, two sequences are either homologous or are not, but cannot be 50% homologous. This term should be changed to “sequence identity” throughout the manuscript.

Done, changed.

-       Comment 3: The terms paralog and isoform are used interchangeably throughout the manuscript, yet they are not equivalent (for example P4, L92; Table 2). Authors should clearly state whether they are reporting paralogs or isoforms, and how are they defining each. There is already variation in the literature when referring to protein isoforms that leads to confusion, and thus authors should clarify this point.

Thank you, "isoform" was changed to "paralog" thoroughly.

-       Comment 4: In the section Transcriptomic intraspecific variation, authors state that they found more isoforms and recovered 1.5 times as many cDNA reads in S2 than in S1 (P4, L96-98). This could be the result of lower integrity and quality of the isolated mRNA and the resulting cDNA libraries from one of the samples, which would also require a more stringent trimming and removal of a greater number of raw reads. Authors should also report the initial number of raw reads per sample and the RNA RIN or other appropriate measurement of RNA quality/integrity measurement.

Fragment analyzer results on both RNA extractions show similar quality, therefore we believe this is natural variation and not due to experimental bias. In addition, we have found similar results in our recently published study on C. tulipa venom(Dutt et al., 2019) and others have also reported significant variations between specimens in other species(Abalde et al., 2018).

-       Comment 5: In the Discussion authors argue (P11, L308) that certain conotoxins previously found in C. imperialis but missing in their study might not be critical components of the venom. I think authors should revise this statement as there are many factors that could account for this discrepancy, for example, these particular conotoxins might be important components in the defensive venom and not in the predatory venom, or as they state in the very next sentence, the differences might also be due to geographical variation, or even size or life stage of the analyzed specimen.

Thank you. The sentence now read as: "suggesting that toxin expression profile might be linked to many factors such as geographical variation..."

Other minor comments:

-       P1, L36 – Missing period after [14]

-       P2, L53 – This is the first time that C. imperialis is mentioned in the text and therefore the genus name should be written in full.

-       The μ sign is missing in several places throughout the manuscript (P12, L363; P13, L399-400).

Thank you. All corrected.

Round 2

Reviewer 4 Report

Dear authors,

Thank you for your response to my comments and for your revisions. In general I am satisfied with the explanations you provided to my questions, and I appreciate your thorough responses.

However, I still think that there is missing information that needs to be included in the manuscript. I think one of the most efficient ways to estimate conotoxin expression levels in a study like yours, would be to map raw reads to a reference genome or transcriptome (options 2a or 2b in your reply), but I understand that generating a full transcriptome with deep coverage might be out of the scope of your study, or maybe you did not have the means to do it. In any case, I still think details regarding the sequencing and transcriptome assembly should be included in the methods. It is important information that the readers need to know to properly evaluate your results.

In addition, you mention that the fragment analyzer results of your RNA extractions show similar results, and thus you believe your results are due to natural variation. I would please ask that you specify this in the text, and include the relevant data supporting your claim in the Supplementary material (quality measurements of RNA and initial numbers of raw reads per sample, reads after trimming, etc.).

Thank you very much and congratulations on your work.

Author Response

Reviewer 4:

Thank you for your response to my comments and for your revisions. In general I am satisfied with the explanations you provided to my questions, and I appreciate your thorough responses. 

Reviewer: However, I still think that there is missing information that needs to be included in the manuscript. I think one of the most efficient ways to estimate conotoxin expression levels in a study like yours, would be to map raw reads to a reference genome or transcriptome (options 2a or 2b in your reply), but I understand that generating a full transcriptome with deep coverage might be out of the scope of your study, or maybe you did not have the means to do it. In any case, I still think details regarding the sequencing and transcriptome assembly should be included in the methods.

Answer: We agree with the reviewer's suggestion, and the following sequencing and transcriptome assembly details have been included in the methods:

Page 6, we added a paragraph in the method section specifically for the assembly procedures (line 422-426):

"Standard Flowgram Format, FASTA and quality file from 454 containing de-multiplexed read sequences were submitted independently to two different de novo assemblers: Newbler (version 2.6) with default parameters (notably: minimum contig length for all contigs = 100, minimum contig length for large contigs = 500, minimum length of reads to use in assembly = 50), and Trinity (version v2.4.0) in its default configuration and with minimum assembled contig length of 100."

Reviewer: It is important information that the readers need to know to properly evaluate your results.  In addition, you mention that the fragment analyzer results of your RNA extractions show similar results, and thus you believe your results are due to natural variation. I would please ask that you specify this in the text,

Answer: First, a comparison Supplementary Table 3 of the sequencing outputs and assembly results has been added in Supplementary material. The original Supplementary Table 3 was changed to 4.

Supplementary material Page 4:

Supplementary Table 3: Assembly of C imperialis venom duct

IMP   S1 (Total De-multiplexed 249,349 and total assembled 198,889)

Raw   Reads

Newbler   Contigs

Newbler   Isotigs

Trinity

#   Sequences

220,516

703

473

1,594

Length   Interval

24-770

1-3,060

58-3,060

101-3,075

Average   Length

396.70

436.81

709.57

481.38

N50

459

700

770

502

N75

504

1,197

1,301

818

N90

528

1,690

1,723

1,387

%GC

52.41

47.67

47.04

46.30

%N

0

0

0

0

Total   # bases

87,478,648

307,076

335,626

767,323

IMP   S2 (Total De-multiplexed 379,998 and total assembled 281,952)   

Raw   Reads

Newbler   Contigs

Newbler   Isotigs

Trinity

#   Sequences

332,390

2,256

1,386

6,021

Length   Interval

22-737

1-6,389

26-6,389

101-6,994

Average   Length

393.38

463.77

816.63

497.22

N50

459

824

953

521

N75

503

1,382

1,619

868

N90

527

2,218

2,513

1,428

%GC

50.62

46.59

46.36

45.34

%N

0

0

0

0

Total   # bases

130,754,372

1,046,266

1,131,845

2,993,735

The following text has been added and modified, lines 69-90 now read as:

"Newbler and Trinity assembled de-multiplexed read sequences that produced contigs with consistent length ranges, N50, N75 and N90 values (Supplementary Table 3). The average length after assembly was not much different compared to the raw reads, e.g. the trimmed raw reads average length ~390 bp, after the Trinity assembly ~480 bp, and after Newbler assembly 430-460 bp.

Sorting and classification into gene superfamilies of the raw cDNA reads was performed using ConoSorter[20]. After motif searching using parameters generated from the ConoServer database[21], a total of 267 unique conopeptide precursors were retrieved including 96 from the specimen S1 and 233 from specimen S2, with 62 overlapping (Figure 1A and Supplementary Table 1). Assembled data sets were also subjected to the same conotoxin sequence analysis, however, the number of conopeptide sequences identified from the assembled contigs with ConoSorter remained low compared to the direct analysis of the cDNA reads. From the 62 common sequences retrieved initially from the cDNA read datasets of both specimens, only 15 (Newbler) and 18 (Trinity) presented after the assembly. Whereas the low expressed sequences have been eliminated during the assembly as expected, it was quite surprising that some highly expressed sequences were absent after the assembly (Supplementary Table 4). Indeed, most algorithms are designed to reduce substitutions, deletions and insertions events, which, in the case of hypervariable genes such as conotoxins, will likely eliminate some minor yet important true biological variations [22]. Comparison of conopeptide sequences measured at the cDNA level allows a clear distinction between high and low levels of expressed transcripts. Therefore, this work from now on was carried out using the reads not contigs as the short sequences of conotoxin precursors (70-100 a.a. on average) has been proven fully covered by most 454 reads (average length 390 bp)."

Reviewer: and include the relevant data supporting your claim in the Supplementary material (quality measurements of RNA and initial numbers of raw reads per sample, reads after trimming, etc.). 

Answer: The report of AGRF RNA quality measurement (2100 expert_mRNA Pico_DE54700697) has been added in Supplementary material P7-9. The initial numbers of raw reads per sample, reads after trimming and used for assembly has been included in Supplementary Table 3.The final submission quantity for sequencing was 200 ng for both samples as specified in text lines 420-421.

Thank you.